# Practicality of generalization guarantees for unsupervised domain adaptation with neural networks

**Adam Breitholtz**                                                 *adambre@chalmers.se*
*Department of Computer Science*
*Chalmers University of Technology*

**Fredrik D. Johansson**                                   *fredrik.johansson@chalmers.se*
*Department of Computer Science*
*Chalmers University of Technology*

**Reviewed on OpenReview:** *https://openreview.net/forum?id=vUuHPRrWs2*

## Abstract

Understanding generalization is crucial to confidently engineer and deploy machine learning models, especially when deployment implies a shift in the data domain. For such domain adaptation problems, we seek generalization bounds which are tractably computable and tight. If these desiderata can be reached, the bounds can serve as guarantees for adequate performance in deployment. However, in applications where deep neural networks are the models of choice, deriving results which fulfill these remains an unresolved challenge; most existing bounds are either vacuous or has non-estimable terms, even in favorable conditions. In this work, we evaluate existing bounds from the literature with potential to satisfy our desiderata on domain adaptation image classification tasks, where deep neural networks are preferred. We find that all bounds are vacuous and that sample generalization terms account for much of the observed looseness, especially when these terms interact with measures of domain shift. To overcome this and arrive at the tightest possible results, we combine each bound with recent data-dependent PAC-Bayes analysis, greatly improving the guarantees. We find that, when domain overlap can be assumed, a simple importance weighting extension of previous work provides the tightest estimable bound. Finally, we study which terms dominate the bounds and identify possible directions for further improvement.

## 1 Introduction

Successful deployment of machine learning systems relies on generalization to inputs never seen in training. In many cases, training and in-deployment inputs differ systematically; these are domain adaptation (DA) problems. An example of a setting where these problems arise is healthcare. Learning a classifier from data from one hospital and applying to samples from another is an example of tasks that machine learning often fail at (AlBadawy et al., 2018; Perone et al., 2019; Castro et al., 2020). In high-stakes settings like healthcare, guarantees on model performance would be required before meaningful deployment is accepted. Modern machine learning models, especially neural networks, perform well on diverse and challenging tasks on which conventional models have had only modest success. However, due to the high flexibility and opaque nature of neural networks, it is often hard to quantify how well we can expect them to perform in practice.

Performance guarantees for machine learning models are typically expressed as generalization bounds. Bounds for unsupervised domain adaptation (UDA), where no labels are available from the target domain, have been explored in a litany of papers using both the PAC (Ben-David et al., 2007; Mansour et al., 2009) and PAC-Bayes (Germain et al., 2020) frameworks; see Redko et al. (2020) for an extensive survey. Despite great interest in this problem, very few works actually compute or report the *value* of the proposed bounds. Instead, the results are used only to guide optimization or algorithm development. Moreover, the bounds

presented often contain terms which are non-estimable without labeled data from the target domain even under favourable conditions (Johansson et al., 2019; Zhao et al., 2019).

For deployment in sensitive settings we wish to find bounds which are: a) Amenable to calculation; they do not contain non-estimable terms and are tractable to compute. b) Tight; they are close to the error in deployment (or at least non-vacuous). How do existing bounds fare in solving this problem? As we will see, for realistic problems under favorable conditions, most, if not all, bounds in the literature struggle to satisfy one or both of these goals to varying degrees.

In this work, we examine the practical usefulness of current UDA bounds as performance guarantees in the context of learning with neural networks. Examining the literature with respect to our desiderata, we identify bounds which show promise in being estimable and tractably computable (Section 2.1). We find that terms related to sample generalization dominate existing bounds for neural networks, prohibiting tight guarantees. To remedy this, we apply PAC-Bayes analysis (McAllester, 1999) with data-dependent priors (Dziugaite & Roy, 2019) in four diverse bounds (Sections 2.3–2.4). Two are existing PAC-Bayes bounds from the UDA literature and two are PAC-Bayes adaptations of bounds based on importance weighting (IW) and integral probability metrics (IPM). We evaluate the bounds empirically under favorable conditions in two tasks which fulfill the covariate shift and domain overlap assumptions; one task concerns digit image classification and the second X-ray classification (Section 3). Our results show that all four bounds are vacuous on both tasks without data-dependent priors, but some can be made tight with them (Section 4). Furthermore, we find that the simple extension of applying importance weights to previous work outperforms the best fully observable bound from the literature in tightness. This result highlights amplification of bound looseness due to interactions between domain adaptation and sample generalization terms. We conclude by offering insights into achieving the tightest bounds possible given the current state of the literature (Section 5).

## 2 Background

In this section, we introduce the unsupervised domain adaptation (UDA) problem and give a survey of existing generalization bounds through the lens of practicality: do the bounds contain non-estimable terms and are they tractably computable? We go on to select a handful of promising bounds and combine them with data-dependent PAC-Bayes analysis to arrive at the tightest guarantees available.

We study UDA for binary classification, in the context of an input space $\mathcal{X} \subseteq \mathbb{R}^d$ and a label space $\mathcal{Y} = \{-1, 1\}$. While our arguments are general, we use as running example the case where $\mathcal{X}$ is a set of black-and-white images. Let $\mathcal{S}$ and $\mathcal{T}$, where $\mathcal{S} \neq \mathcal{T}$, be two distributions, or *domains*, over the product space $\mathcal{X} \times \mathcal{Y}$, called the source domain and target domain respectively. The source domain is observed through a labeled sample $S = \{x_i, y_i\}_{i=1}^n \sim (\mathcal{S})^n$ and the target domain through a sample $S'_x = \{x'_i\}_{i=1}^m \sim (\mathcal{T}_x)^m$ which lacks labels, where $\mathcal{T}_x$ is the marginal distribution on $\mathcal{X}$ under $\mathcal{T}$. Throughout, $(\mathcal{D})^N$ denotes the distribution of a sample of $N$ datapoints drawn i.i.d. from the domain $\mathcal{D}$.

The UDA problem is to learn hypotheses $h$ from a hypothesis class $\mathcal{H}$, by training on $S$ and $S'_x$, such that the hypotheses perform well on unseen data drawn from $\mathcal{T}_x$. In the Bayesian setting, we learn posterior distributions $\rho$ over $\mathcal{H}$ from which sampled hypotheses perform well on average. We measure performance using the expected *target risk* $R_\mathcal{T}$ of a single hypothesis $h$ or posterior $\rho$,

$$\underbrace{R_\mathcal{T}(h) = \mathbb{E}_{(x,y)\sim\mathcal{T}}[\ell(h(x), y)]}_{\text{Risk for single hypothesis } h} \quad \text{or} \quad \underbrace{\mathbb{E}_{h\sim\rho} R_\mathcal{T}(h)}_{\text{Gibbs risk of posterior } \rho} , \tag{1}$$

for a loss function $\ell : \mathcal{Y} \times \mathcal{Y} \to \mathbb{R}_+$. In this work, we study the zero-one loss, $\ell(y, y') = \mathbb{1}[y \neq y']$. The Gibbs risk is used in the PAC-Bayes guarantees (Shawe-Taylor & Williamson, 1997; McAllester, 1998), a generalization of the PAC framework (Valiant, 1984; Vapnik, 1998).

When learning from samples, the empirical risk $\hat{R}_\mathcal{D}$ can be used as an observable measure of performance,

$$\hat{R}_\mathcal{D}(h) = \frac{1}{m} \sum_{i=1}^m \ell(h(x_i), y_i), \tag{2}$$

for a sample $\{(x_i, y_i)\}_{i=1}^n \sim (\mathcal{D})^n$, with the empirical risk of the Gibbs classifier defined analogously. However, since no labeled sample from $\mathcal{T}$ is available, the risk of interest is not directly observable. Hence, the most common way to approximate this quantity is to derive an upper bound on the target risk. We refer to these as *UDA bounds*. Crucially, *any practical performance guarantee must be made using only observed data and assumptions on how $\mathcal{S}$ and $\mathcal{T}$ relate.* Throughout, we make the following common assumptions.

**Assumption 1** (Covariate shift & overlap)**.** The source domain $\mathcal{S}$ and target domain $\mathcal{T}$ satisfy for all $x, y$

$$\textbf{Covariate shift:} \quad \mathcal{T}_y(Y \mid X = x) = \mathcal{S}_y(Y \mid X = x) \quad \text{and} \quad \mathcal{T}_x(X = x) \neq \mathcal{S}_x(X = x)$$
$$\textbf{Overlap:} \quad \mathcal{T}_x(X = x) > 0 \Rightarrow \mathcal{S}_x(X = x) > 0 \ .$$

These are strong assumptions and covariate shift cannot be verified statistically. They are not required by every bound in the literature but together they are sufficient to guarantee identification and consistent estimation of the target risk (Shimodaira, 2000). More importantly, the generalization guarantees we study more closely are not fully observable without them unless target labels are available. Finally, this setting is among the most simple and favorable ones for UDA which should make for an interesting benchmark—if existing bounds are vacuous also here, significant challenges remain.

## 2.1 Overview of existing UDA bounds

Most existing UDA bounds on the target risk share a common structure due to their derivation. The typical process starts by bounding the *expected* target risk using the *expected* source risk and measures of domain shift. Thereafter, terms are added which bound the sample generalization error, the difference between the expected source risk and its empirical estimate. The results can be summarized conceptually, with $f$ and arguments variously defined, as expressions on the form

$$R_\mathcal{T} \leq f(\text{Empirical source risk, Measures of domain shift, Sample generalization error}) \ .$$

There are two main forms taken by this function; one in which sample generalization terms are related to domain shift terms through addition and one where they are multiplied. We call these additive and multiplicative bounds respectively. One example is the classical result due to Ben-David et al. (2007) which uses the so-called $\mathcal{A}$-distance to bound the target risk of $h \in \mathcal{H}$ with probability $\geq 1 - \delta$,

$$R_\mathcal{T}(h) \leq \underbrace{\hat{R}_\mathcal{S}(h)}_{\text{Emp. risk}} + \underbrace{\sqrt{\frac{4(d \log \frac{2em}{d} + \log \frac{4}{\delta})}{m}}}_{\text{Sample generalization}} + \underbrace{d_\mathcal{H}(\mathcal{S}, \mathcal{T}) + \lambda}_{\text{Domain shift}}, \tag{3}$$

where $d$ is the VC dimension of the $\mathcal{H}$, $\lambda$ is the sum of the errors on both domains of the best performing classifier $h^* = \arg\min_{h \in \mathcal{H}}(R_\mathcal{S}(h) + R_\mathcal{T}(h))$, and $d_\mathcal{H}(\mathcal{S}, \mathcal{T}) = 2 \sup_{A \in \{\{x : h(x) = 1\} : h \in \mathcal{H}\}} |\mathrm{Pr}_\mathcal{S}[A] - \mathrm{Pr}_\mathcal{T}[A]|$ is the $\mathcal{A}$-distance for the characteristic sets of hypotheses in $\mathcal{H}$.

Three challenges limits the practicality of this bound: i) $\lambda$ is not directly estimable without target labels and must be assumed small for an informative bound. This is a pattern in UDA theory which illustrates a fundamental link between estimability and assumption. ii) The VC dimension can easily lead to a vacuous result for modern neural networks. For example, the VC dimension of piecewise polynomial networks is $\Omega(pl \log \frac{p}{l})$ where $p$ is the number of parameters and $l$ is the number of layers (Bartlett et al., 2019). iii) The $\mathcal{A}$-distance can be tractably computed only for restricted hypothesis classes. These issues are not unique the bound above, they are exhibited to varying degrees by any UDA bound.

In response to concerns for practical generalization bounds in deep learning, Valle-Pérez & Louis (2020) put forth seven desiderata for predictive bounds. While these are of interest also for UDA, we concern ourselves primarily with the fifth (non-vacuity of the bound) and sixth (efficient computability) desiderata as they are of paramount importance to achieving practically useful guarantees. In this work, we study UDA bounds with emphasis on how each term influences the following properties when learning with deep neural networks.

1. **Tightness.** Is the term a poor approximation? Is it likely to lead to a loose bound?

Table 1: Overview of existing UDA bounds with respect to a) measures of domain divergence, b) whether domain and sample generalization terms add or multiply, c) non-estimable terms, d) whether the bounds were computed empirically, e) computational tractability. The highlighted rows represent a selection of bounds which are possible to estimate under the assumptions we make and which hold promise in fulfilling our other desiderata. $\mathsf{X}^*$ denotes that under the assumptions made in this work we do not have non-estimable terms.

| Paper/reference | Divergence | Add/Mult | Non-est. terms | Evaluated bound | Tractable |
|---|---|---|---|---|---|
| Ben-David et al. (2007) | $\mathcal{A}$-distance | Add | ✓ | X | X |
| Blitzer et al. (2008) | $H\Delta H$ | Add | ✓ | X | X |
| Ben-David et al. (2010) | $H\Delta H$ | Add | ✓ | X | X |
| Morvant et al. (2012) | $H\Delta H$ | Add | ✓ | X | X |
| Mansour et al. (2009) | Discrepancy dist. | Add | ✓ | X | X |
| Redko et al. (2019) | Discrepancy dist. | Add | ✓ | X | X |
| Kuroki et al. (2019) | S-discrepancy | Add | ✓ | X | X |
| Cortes & Mohri (2014) | Gen. discrepancy | Add | ✓ | X | X |
| Cortes et al. (2015) | Gen. discrepancy | Add | ✓ | X | X |
| Zhang et al. (2012) | IPM | Add | X | X | ✓ |
| Redko (2015) | IPM, MMD | Add | ✓ | X | ✓ |
| Long et al. (2015) | MMD | Add | ✓ | X | ✓ |
| Redko et al. (2017) | IPM | Add | ✓ | X | ✓ |
| Johansson et al. (2019) | IPM | Add | ✓ | X | ✓ |
| Zhang et al. (2019) | Margin disparity | Add | ✓ | X | X |
| Dhouib et al. (2020) | Wasserstein | Add | ✓ | X | ✓ |
| Shen et al. (2018) | Wasserstein | Add | ✓ | X | ✓ |
| Courty et al. (2017b) | Wasserstein | Add | ✓ | X | ✓ |
| Germain et al. (2013) | Domain disagreement | Add | ✓ | X | X |
| Zhang et al. (2020) | Localized discrepancy | Add | ✓ | X | X |
| Cortes et al. (2019) | Localized discrepancy | Add | ✓ | X | X |
| Acuna et al. (2021) | f-divergences | Add | ✓ | X | X |
| Germain et al. (2016) | $\beta$-divergence | Mult | $\mathsf{X}^*$ | X | ✓ |
| Cortes et al. (2010) | Rényi | Mult | $\mathsf{X}^*$ | X | X |
| Dhouib & Redko (2018) | $L^1, \chi^2$ | Mult | ✓ | X | X |

2. **Estimability.** Is the term something which we can estimate from observed data?

3. **Computability.** Can we tractably compute it for real-world data sets and hypothesis classes?

Next, we give a short summary of existing UDA bounds, with the excellent survey by Redko et al. (2019) as starting point, while evaluating whether they contain non-estimable terms, if the bound was computed in the paper[1] and if the bound is computationally tractable for neural networks. We have listed considered bounds in Table 1.

We will now reason about the bounds' potential to reach our stated desiderata, starting with tractability. We begin by noting that several divergence measures (e.g., $\mathcal{A}$-distance, $H\Delta H$-distance, Discrepancy distance) are defined as suprema over the hypothesis class $\mathcal{H}$. Typically, these are intractable to compute for neural nets due to the richness of the class, and approximations would yield lower bounds rather than upper bounds. Several works fail to yield practically computable bounds for neural networks for this reason (Ben-David et al., 2007; Blitzer et al., 2008; Ben-David et al., 2010; Morvant et al., 2012; Mansour et al., 2009; Redko et al., 2019; Cortes & Mohri, 2014; Cortes et al., 2015). There are also some works which deal with so called localized discrepancy which depends on finding a subset of promising classifiers and bounding their performance instead. (Zhang et al., 2020; Cortes et al., 2019) However, this subset is not easy to find in general and as such we view these approaches as intractable also.

---

[1] By this we mean the whole bound. Some of the works listed have computed one or several parts of their bound. However, the computation is generally done for simpler model classes than neural networks.

Continuing with estimability, we may remove from consideration also those whose non-estimable term cannot be dealt with without assuming that they are small—an untestable assumption which does not follow from overlap and covariate shift. This immediately disqualifies a large swathe of bounds which all include the joint error of the optimal hypothesis on both domains, or some version thereof, a very common non-estimable term in DA bounds (Kuroki et al., 2019; Redko, 2015; Long et al., 2015; Redko et al., 2017; Johansson et al., 2019; Zhang et al., 2019; Dhouib et al., 2020; Shen et al., 2018; Courty et al., 2017b; Germain et al., 2013; Dhouib & Redko, 2018; Acuna et al., 2021). In principle, we might be able to approximate this quantity under overlap, e.g. by using importance sampling. However, this would entail solving a new optimization problem to find a hypothesis which has a low joint error (see discussion after equation 3). If we instead wish to upper bound the term, we must solve an equivalent problem to the one we are trying to solve in the first place.

Three bounds remain: Zhang et al. (2012) use integral probability metrics (IPM) between source and target domains to account for covariate shift in their bound, which are tractable to compute under assumptions on the hypothesis class; Cortes et al. (2010) use importance weights (IW) and Renyi divergences which are well-defined and easily computed under overlap; Germain et al. (2016) use a related metric based on the norm of the density ratio between domains with similar properties. Respectively, the first two results bound sample generalization error using the uniform entropy number and the covering number. These measures are intractable to compute for neural networks, and while they may be upper bounded by the VC dimension (Wainwright, 2019), this is typically large enough to yield uninformative guarantees (Zhang et al., 2021). In contrast, Germain et al. (2016) use a PAC-Bayes analysis which we can apply to neural networks by specifying prior and posterior distributions over network weights. Using Gaussian distributions for both priors and posteriors, the PAC-Bayes bound can be readily computed. Thus, to enable closer comparison and tractable computation, in the coming sections, we unify each bounds' dependence on sample generalization by adapting IW and IPM bounds to the PAC-Bayes framework. For completeness, we include an additional PAC-Bayes bound from the UDA literature, which has a non-estimable term, namely Germain et al. (2013).

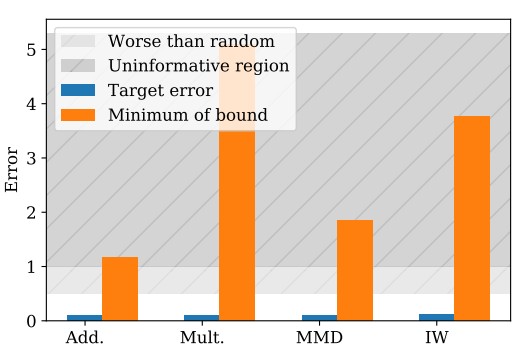

Figure 1: Best bounds achieved without data-dependent priors on the MNIST/MNIST-M task using LeNet-5 as well as the target error for the same posterior hypothesis. Note that all the bounds are vacuous, i.e. they are above one.

The selected bounds are given in Sections 2.3–2.4. Germain et al. (2016) will be referred to as the multiplicative bound (**Mult**), Germain et al. (2013) as the additive bound (**Add**), the adaptation of importance weighting to PAC-Bayes as **IW**; and the adaptation of IPM bounds as **MMD**. Unfortunately, also the resulting PAC-Bayes bounds are uninformative for even simple image classification UDA tasks; see Figure 1. Consistent with our understanding of standard learning with neural networks, *we find both classical PAC and PAC-Bayes bounds vacuous in the tasks most frequently used as empirical benchmarks in papers deriving UDA bounds.* Next, we detail how to make use of data-dependent priors to get the tightest possible bounds.

## 2.2 Tighter sample generalization guarantees using PAC-Bayes with data-dependent priors

PAC-Bayes theory studies generalization of a posterior distribution $\rho$ over hypotheses in $\mathcal{H}$, learned from data, in the context of a prior distribution over hypotheses, $\pi$. The generalization error in $\rho$ may be bounded using the divergence between $\rho$ and $\pi$ as seen in the following classical result due to McAllester.

**Theorem 1** (Adapted from Thm. 2 in McAllester (2013)). *For a prior $\pi$ and posterior $\rho$ on $\mathcal{H}$, a bounded loss function $\ell : \mathcal{Y} \times \mathcal{Y} \to [0, 1]$ and any fixed $\gamma, \delta \in (0, 1)$, we have w.p. at least $1 - \delta$ over the draw of $m$*

samples from $\mathcal{D}$, with $D_{\mathrm{KL}}(p\|q)$ the Kullback-Liebler (KL) divergence between $p$ and $q$,

$$\mathbb{E}_{h\sim\rho} R_{\mathcal{D}}(h) \leq \frac{1}{\gamma} \mathbb{E}_{h\sim\rho} \hat{R}_{\mathcal{D}}(h) + \frac{D_{\mathrm{KL}}(\rho\|\pi) + \ln(\frac{1}{\delta})}{2\gamma(1-\gamma)m} \; .$$

The bound in Theorem 1 grows loose when prior $\pi$ and posterior $\rho$ diverge—when the posterior is sensitive to the training data. When learning with neural networks, $\pi$ and $\rho$ are typically taken to be distributions on the weights of the network before and after training. However, the weights of a trained deep neural network will be far away from any uninformed prior after only a few epochs. For this reason, Dziugaite et al. (2021) developed a methodology, based on work by Ambroladze et al. (2007) and Parrado-Hernández et al. (2012), for learning *data-dependent* neural network priors by a clever use of sample splitting. To ensure that the bound remains valid, any data which is used to fit the prior must be independent of the data used to evaluate the bound. In this work, we learn $\pi$ and $\rho$ following Dziugaite et al. (2021), as described below.

1. A fraction $\alpha \in [0,1)$ is chosen and the available training data, $S$, is split randomly into two parts, $S_\alpha$ and $S \setminus S_\alpha$ of size $\alpha m$ and $(1-\alpha)m$, respectively.

2. A neural network is randomly initialized and trained using stochastic gradient descent on $S_\alpha$ for one epoch. From this we get the weights, $w_\alpha$.

3. The same network is trained, starting from $w_\alpha$, on all of $S$ until a stopping condition is satisfied. In this work, we terminate training after 5 epochs. We save the weights, $w_\rho$.

4. From $w_\alpha$ and $w_\rho$ we create our prior and posterior from Normal distributions centered on the learned weights, $\pi = \mathcal{N}(w_\alpha, \sigma I)$ and $\rho = \mathcal{N}(w_\rho, \sigma I)$ respectively. $\sigma$ is a hyperparameter governing the specificity (variance) of the prior which may be chosen when evaluating.

5. Finally, we use the learned prior and posterior to evaluate the bounds on $S \setminus S_\alpha$.

### 2.3 PAC-Bayes bounds from the domain adaptation literature

Both the additive and multiplicative PAC-Bayes UDA bounds described in Section 2.1 are defined in Germain et al. (2020) and make use of a decomposition of the zero-one risk into the *expected joint error*

$$e_{\mathcal{D}}(\rho) = \mathbb{E}_{h,h'\sim\rho\times\rho} \mathbb{E}_{x,y\sim\mathcal{D}} \ell(h(x), y)\ell(h'(x), y),$$

which measures how often two classifiers drawn from $\rho$ make the same errors, and the *expected disagreement*

$$d_{\mathcal{D}_x}(\rho) = \mathbb{E}_{h,h'\sim\rho\times\rho} \mathbb{E}_{x\sim\mathcal{D}_x} \ell(h(x), h'(x)),$$

which measures how often two classifier disagree on the labeling of the same point. Empirical variants $\hat{e}_{\mathcal{D}}(\rho)$ and $\hat{d}_{\mathcal{D}_x}(\rho)$ replace expectations with sample averages analogous to equation 2.

In the bound of Germain et al. (2016), the sample generalization component (KL-divergence between prior and posterior) is multiplied with a domain shift component (supremum density ratio).

**Theorem 2** (Multiplicative bound, Germain et al. (2016)). *For any real numbers $a, b > 0$ and $\delta \in (0,1)$, it holds, under Assumption 1, with probability at least $1-\delta$ over labeled source samples $S \sim (\mathcal{S})^m$ and unlabeled target samples $T_x \sim (\mathcal{T}_x)^n$, with constants $a' = \frac{a}{1-e^{-a}}$, $b' = \frac{b}{1-e^{-b}}$, for every posterior $\rho$ on $\mathcal{H}$ that*

$$\mathbb{E}_{h\sim\rho} R_{\mathcal{T}}(h) \leq a'\frac{1}{2}\hat{d}_{\mathcal{T}_x} + b'\beta_\infty(\mathcal{T}\|\mathcal{S})\hat{e}_{\mathcal{S}} + \left(\frac{a'}{na} + \frac{b'\beta_\infty(\mathcal{T}\|\mathcal{S})}{mb}\right)\left(2D_{\mathrm{KL}}(\rho\|\pi) + \ln\frac{2}{\delta}\right) + \eta_{\mathcal{T}\setminus\mathcal{S}} \; ,$$

*with $\beta_\infty(\mathcal{T}\|\mathcal{S}) = \sup_{x\in\mathrm{supp}(\mathcal{S}_x)} \mathcal{T}_x(x)/\mathcal{S}_x(x),$[2] and $\eta_{\mathcal{T}\setminus\mathcal{S}} = \mathbb{E}_{(x,y)\sim\mathcal{T}}[\mathbb{1}[(x,y) \notin \mathrm{supp}(\mathcal{S})]] \sup_{h\in\mathcal{H}} R_{\mathcal{T}\setminus\mathcal{S}}(h).$*

---

[2]Terms $\beta_q$ based on the $q$:th moment of the density ratio for $q < \infty$ are considered in (Germain et al., 2020) but not here.

The bound is simplified slightly in our setting due to Assumption 1 as $\eta_{\mathcal{T}\setminus\mathcal{S}}$ will be 0. By Bayes rule, $\beta_\infty$ can be computed as the maximum ratio between conditional probabilities of an input being sampled from $\mathcal{T}$ or $\mathcal{S}$. The second result due to Germain et al. is additive in its interaction between domain and sample generalization terms.

**Theorem 3** (Additive bound, Germain et al. (2013))**.** *For any real numbers $\omega, \gamma > 0$ and $\delta \in (0,1)$, with probability at least $1 - \delta$ over labeled source samples $S \sim (\mathcal{S})^m$ and unlabeled target samples $T_x \sim (\mathcal{T}_x)^m$; for every posterior $\rho$ on $\mathcal{H}$, it holds with constants $\omega' = \frac{\omega}{1-e^{-\omega}}$ and $\gamma' = \frac{2\gamma}{1-e^{-2\gamma}}$ that*

$$\underset{h\sim\rho}{\mathbb{E}} R_\mathcal{T}(h) \leq \underset{h\sim\rho}{\mathbb{E}} \omega' \hat{R}_\mathcal{S}(h) + \gamma' \frac{1}{2} \hat{Dis}_\rho(S, T_x) + \left(\frac{\omega'}{\omega} + \frac{\gamma'}{\gamma}\right) \frac{D_{\mathrm{KL}}(\rho\|\pi) + \log\frac{3}{\delta}}{m} + \lambda_\rho + \frac{1}{2}(\gamma' - 1),$$

*where $\hat{Dis}_\rho(S, T_x) = |\hat{d}_{\mathcal{T}_x} - \hat{d}_{\mathcal{S}_x}|$ is the empirical domain disagreement, $\lambda_\rho = |e_\mathcal{T}(\rho) - e_\mathcal{S}(\rho)|$.*

Next we will combine the main techniques used to account for domain shift in Cortes et al. (2010) and Zhang et al. (2012) with the PAC-Bayes analysis of Theorem 1, producing two corollaries for the UDA setting.

### 2.4 Adapting the classical PAC-Bayes bound to unsupervised domain adaptation

First, we adapt the bound in Theorem 1 to UDA by incorporating importance weighting (Shimodaira, 2000; Cortes et al., 2010).We define a weighted loss $\ell^w(h(x), y) = w(x)\ell(h(x), y)$ where $w(x) = \frac{T(x)}{S(x)}$ and $\ell$ is the zero-one loss. The risk of a hypothesis using this loss is denoted by $R^w$.

**Corollary 1.** *(IW bound) Consider the conditions in Theorem 1 and let $\beta_\infty = \sup_{x\sim\mathcal{X}} w(x)$. We have, for any choice of $\gamma, \delta \in (0,1)$ and any pick of prior $\pi$ and posterior $\rho$ on $\mathcal{H}$,*

$$\underset{h\sim\rho}{\mathbb{E}} R_\mathcal{T}(h) \leq \frac{1}{\gamma}\underset{h\sim\rho}{\mathbb{E}} \hat{R}_\mathcal{S}^w(h) + \beta_\infty \frac{D_{\mathrm{KL}}(\rho\|\pi) + \ln(\frac{1}{\delta})}{2\gamma(1-\gamma)m} .$$

*Proof.* Since Theorem 1 holds for loss functions mapping onto $[0, 1]$, we divide the weighted loss $\ell^w$ by the maximum weight, $\beta_\infty$. The argument then follows naturally when we apply Theorem 1 with the loss function $\frac{\ell^w}{w_{max}}$.

$$\underset{h\sim\rho,(x,y)\sim\mathcal{T}}{\mathbb{E}} \left[\frac{\ell(h(x),y)}{\beta_\infty}\right] = \underset{h\sim\rho,(x,y)\sim\mathcal{S}}{\mathbb{E}} \left[\frac{\ell^w(h(x),y)}{\beta_\infty}\right] \leq \frac{1}{\gamma\beta_\infty}\hat{R}_\mathcal{S}^w + \frac{D_{\mathrm{KL}}(\rho\|\pi) + \ln(\frac{1}{\delta})}{2\gamma(1-\gamma)m}$$

The first equality holds due to Assumption 1 and the definitions of $w$ and $\ell^w$. $\qquad\square$

Now we continue with applying an argument based on integral probability metrics (IPM) drawn from Zhang et al. (2012) and similar works. IPMs, such as the kernel maximum mean discrepancy (MMD) (Gretton et al., 2012) and the Wasserstein distance have been used to give tractable and even differentiable bounds on target error in UDA to guide algorithm development (Courty et al., 2017a; Long et al., 2015). The kernel MMD is a IPM, defined as follows, in terms of its square, given a reproducing kernel $k(\cdot, \cdot) : \mathcal{X} \times \mathcal{X} \to \mathbb{R}$

$$\mathbf{MMD}_k(P, Q)^2 = \underset{X\sim P, X'\sim P}{\mathbb{E}}[k(X, X')] - 2\underset{X\sim p, Y\sim Q}{\mathbb{E}}[k(X, Y)] + \underset{Y\sim Q, Y'\sim Q}{\mathbb{E}}[k(Y, Y')].$$

Here, $X$ and $Y$ are random variables, and $X'$ is an independent copy of $X$ with the same distribution and $Y'$ is an independent copy of $Y$. This measures a notion of discrepancy between the distributions $P$ and $Q$ based on their samples.

We combine the MMD with the bound from McAllester (2013) to arrive at what we will call the **MMD** bound.

**Corollary 2.** *(MMD bound) Let $\bar{\ell}_h(x) = \mathbb{E}[\ell(h(x), Y) \mid X = x]$ be the expected pointwise loss at $x \in \mathcal{X}$ and assume that, for any $h \in \mathcal{H}$, $\bar{\ell}_h$ can be uniformly bounded by a function in reproducing-kernel Hilbert $\mathcal{L}$ space with kernel $k$ such that $\forall x, x' \in \mathcal{X} : 0 \leq k(x, x') \leq K$. Then, under Assumption 1, with $\gamma, \delta \in (0, 1)$ and probability $\geq 1 - \delta$ over labeled source samples $S \sim (\mathcal{S})^m$ and unlabeled target samples $S'_x \sim \mathcal{T}_x^m$,*

$$\underset{h\sim\rho}{\mathbb{E}} R_\mathcal{T}(h) \leq \frac{1}{\gamma}\underset{h\sim\rho}{\mathbb{E}} \hat{R}_\mathcal{S}(h) + \frac{D_{\mathrm{KL}}(\rho\|\pi) + \log\frac{2}{\delta}}{2\gamma(1-\gamma)m} + \hat{\mathrm{MMD}}_k(\mathcal{S}, \mathcal{T}) + 2\sqrt{\frac{K}{m}}\left(2 + \sqrt{\log\frac{4}{\delta}}\right), \tag{4}$$

where $\hat{\mathrm{MMD}}_k(\mathcal{S}, \mathcal{T})$ *is the biased empirical estimate of the maximum mean discrepancy between $\mathcal{S}$ and $\mathcal{T}$ computed from $S_x$ and $S'_x$, see eq. (2) in Gretton et al. (2012).*

*Proof.* By assumption, for any hypothesis $h \in \mathcal{H}$,

$$R_{\mathcal{T}}(h) = R_{\mathcal{S}}(h) + \mathbb{E}_{\mathcal{T}}[\ell(h(X), Y)] - \mathbb{E}_{\mathcal{S}}[\ell(h(X), Y)] \leq R_{\mathcal{S}}(h) + \sup_{l \in \mathcal{L}} |\mathbb{E}_{\mathcal{T}_x}[l_h(X)] - \mathbb{E}_{\mathcal{S}_x}[l_h(X)]| .$$

The inequality holds because $\mathbb{E}_{\mathcal{S}}[\ell(h(x), Y) \mid X = x] = \mathbb{E}_{\mathcal{T}}[\ell(h(x), Y) \mid X = x]$ due to Assumption 1 (covariate shift) and the assumption that $\ell_h$ is uniformly bounded by a function in $\mathcal{L}$. The RHS of the inequality is precisely $\mathrm{MMD}_{\mathcal{L}}$ and the full result follows by linearity of expectation (over $\rho$), since the MMD term is independent of $h$, and application of Theorem 1. The right-most term in equation 4 follows from a finite-sample bound on MMD, Theorem 7 in Gretton et al. (2012), and a union bound w.r.t. $\delta$. $\qquad \square$

**Representation learning.** When no function family $\mathcal{L}$ satisfying the conditions of Corollary 2 is known, an additional unobservable error term must be added to the bound, to account for excess error. Bounds based on the MMD and other IPMs have been used heuristically in representation learning to find representations which minimize the induced distance between domains and achieve better domain adaptation (Long et al., 2015). However, even if covariate shift holds in the input space, these are not guaranteed to hold in the learned representation. For this reason, we do not explore such approaches even though they might hold some promise. An example of this is recent work by Wu et al. (2019) which provided some interesting ideas about assumptions which constrains the structure of the source and target distributions under a specific representation mapping. Exploring such ideas further is an interesting direction for future work.

## 3 Experimental setup

We describe the experimental setup briefly, leaving more details in Appendix A. We examine the dynamics of the chosen bounds (**Add**, **Mult**, **MMD**, **IW**), which parts of the bounds dominate their value, the effect of varying the amount of data used to inform the prior and if the bounds have utility for early stopping indication or model selection. In addition to these we also want to answer if these bounds practically useful as guarantees, and if not, what is lacking and what future directions should be explored to reach our desiderata? Since the term $\lambda_\rho$ in **Add** (Theorem 3) depends on target labels, we give access to these for purposes of analysis and illustration, keeping in mind that the bound is not fully estimable in practice.

We perform experiments on two image classification tasks, described further below, using three different neural network architectures. The architectures used are: a modified version of LeNet-5 due to Zhou et al. (2019), a fully connected network and a version of ResNet50 (He et al., 2016). These specific architectures were picked as examples due to their varying parameter size and complexity. The experiments are repeated for 5 distinct random seeds, with the exception of the varying of image size where we only conduct the experiment for one seed.

We learn prior and posterior network weights as described in Section 2.2. When both sets of weights have been trained, we use these as the means for prior and posterior distributions $\pi$ and $\rho$, chosen to be isotropic Gaussians with equal variance, $\sigma$. Each bound is calculated for each pair of prior and posterior. To estimate the expectation over the posterior we sample 5 pairs of classifiers from the posterior and average their bound components (e.g., source risk) to get an estimate of the bound parts. When this has been calculated, we perform a small optimization step to choose the free parameters of the different bound through a simple grid search over a small number of combinations. We use the combination that produces the lowest minimum bound and account for testing all $k$ combinations by applying a union bound argument, modifying the certainty parameter $\delta$ to $\delta/k$. When calculating the MMD, we use the linear statistic for the MMD as detailed in Gretton et al. (2012) with the kernel $k(x, y) = \exp(\frac{-\|x - y\|^2}{2\kappa^2})$. This calculation is averaged over 10 random shuffles of the data for a chosen bandwidth, $\kappa > 0$. The process is repeated for different choices of bandwidth (see Appendix for details) and the maximum of the results is taken as the value of the MMD. Note that we calculate the MMD in the input space and have not adjusted it for sample variance. When calculating the importance weights we assume here that the maximum weight, $\beta_\infty$, is uniformly bounded as

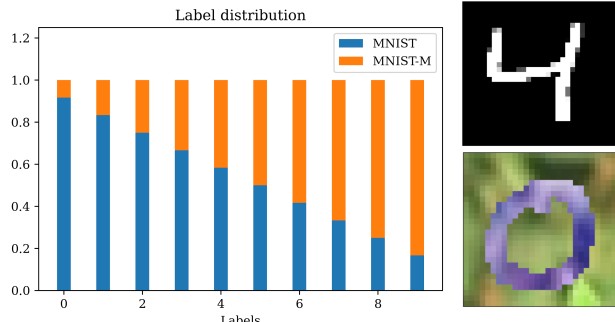

Figure 2: Example of source label densities in the task with the mix of MNIST and MNIST-M samples. An example from each of the two data sets can be seen on the right, the upper one being from MNIST and lower one from MNIST-M. The target domain is the complement of source samples.

the bound would potentially be vacuous otherwise. In addition, we will consider the importance weights to be perfectly computed for simplicity.

We construct two tasks from standard data sets which both fulfill Assumption 1 by design, one based on digit classification and one real-world task involving classification of X-ray images. These are meant to represent realistic UDA tasks where neural networks are the model class of choice.

### 3.1 Task 1: MNIST mixture

MNIST (Lecun, 1998) is a digit classification data set containing 70000 images widely used as a benchmark for image classifiers. MNIST-M was introduced by Ganin et al. (2016) to study domain adaptation and is a variation of MNIST where the digits have been blended with patches taken from images from the BSDS500 data set (Arbeláez et al., 2011). We use MNIST and MNIST-M to construct source and target domains, both of which contain samples from each data set, but with different label density for images from MNIST and MNIST-M. To create the source data set, we start with images labeled "0" by adding 1/12th of the samples from MNIST-M and 11/12th of the samples from MNIST, we increase the proportion from MNIST-M by 1/12 for each subsequent label, "1", "2", and so on. The complement of the source samples is then used as the target data, see Figure 2 for an illustration. We make this into a binary classification problem by relabeling digits 0-4 to "0" and the rest to "1". The supremum density ratio $\beta_\infty \approx 11$ (see Theorem 2) is known and the mixture guarantees overlap in the support of the domains (Assumption 1).

### 3.2 Task 2: X-ray mixture

ChestX-ray14 (Wang et al., 2017) and CheXpert (Irvin et al., 2019) are data sets of chest X-rays and labeled according to the presence of common thorax diseases. The data sets contain 112,120 and 224,316 labeled images, respectively. Since the two data sets do not have full overlap in labels, we use the subset of labels for which there is overlap. The labels which occur in both data sets are: No finding, Cardiomegaly, Edema, Consolidation, Atelectasis, and Pleural Effusion. In addition, there is an uncertainty parameter present in the CheXpert data set which indicates how certain a label is. As we consider binary classification, we set all labels that are uncertain to positive. Therefore, a single image in the CheXpert data set might have multiple associated labels. For most experiments we resize all the images in the data sets to 32x32 to be able to use the same architectures for both tasks. However, we also conduct a small experiment with ResNet50 where larger image sizes are considered.

We take 20% of chestX-ray14 and add it to CheXpert to create our source data set. The target is the remaining part of ChestX-ray14 which is then 89,696 images compared to the 246,072 of the source. With this, we know from Appendix B that $\beta_\infty \approx 11$ and we have overlap. We turn this into a binary classification problem in a one-vs-rest fashion by picking one specific label to identify, relabeling images with "1" if it is

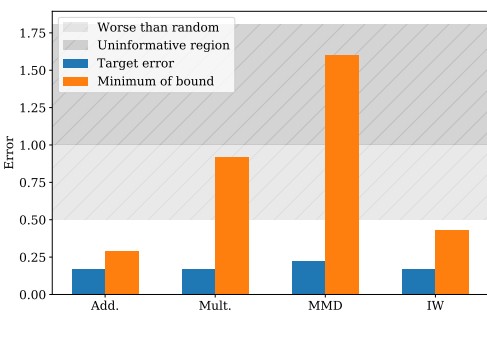

(a) MNIST/MNIST-M task.

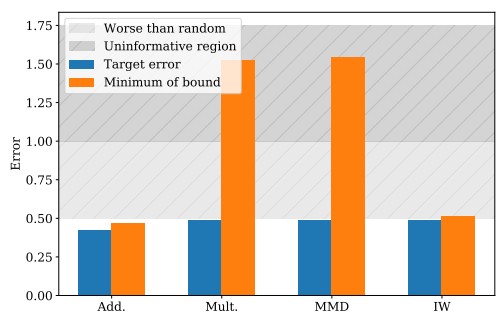

(b) CheXpert+ChestX-ray14 task

Figure 3: The tightest bounds achieved on the LeNet-5 architecture. This illustrates the tightening effect of using data-dependent priors. Non-vacuous bounds are obtained when using data to inform the prior. The shaded area between 0.5 and 1 is where a random classifier would perform on average, and the shaded area above 1 signifies vacuity.

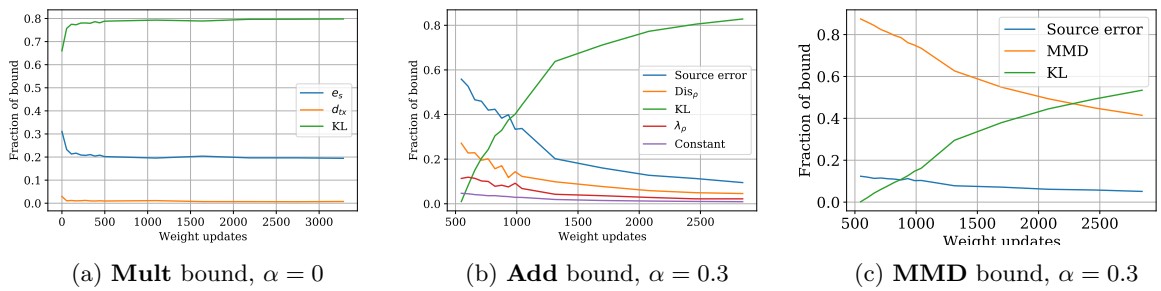

(a) **Mult** bound, $\alpha = 0$      (b) **Add** bound, $\alpha = 0.3$      (c) **MMD** bound, $\alpha = 0.3$

Figure 4: An illustration of constituent parts for three of the bounds with the fully connected architecture on the MNIST mixture task. $\sigma = 0.03$

present, and setting the labels of images with any other finding to "0". In this work, we consider only the task of classifying "No Finding".

## 4 Results

As expected, when we compute bounds with data-dependent priors, we achieve bounds which are substantially tighter than without them, as seen clearly by comparing Figure 1 to Figure 3. We also observe that the additive bound (**Add**) due to Germain et al. (2013) is the tightest overall for both tasks, followed closely by the **IW** bound. The latter is not so surprising as when we apply data-dependent priors, there is effectively a point in training where the $D_{\mathrm{KL}}$-divergence between prior and posterior networks is very small. Moreover, due to overlap, the weighted source error is equal to the target error in expectation. Thus the only sources of looseness left is the error in the approximation of the expectation over the posterior and the $\log \frac{1}{\delta}$ term which is very small here. We can also see in Figure 5a and 5b that the minimum of the **IW** bound is often very close to the minimum of the additive bound. However, unlike **IW**, the **Add** bound relies on access to target labels in to compute the term $\lambda_\rho$ (see further discussion below).

The evolution of the different bounds during training is shown for both tasks in Figure 6. Of course, all bounds will increase at some point as training progresses and the prior and posterior diverges further from each other and $D_{\mathrm{KL}}$ increases. While **Add** is consistently very tight, we note that the $\lambda_\rho$ term which we cannot observe might be a significant part of the bound when the $D_{\mathrm{KL}}$-term is low as we can see in Figure 4b. This is an issue for the additive bound since if we have sufficiently small variance of the posterior then the

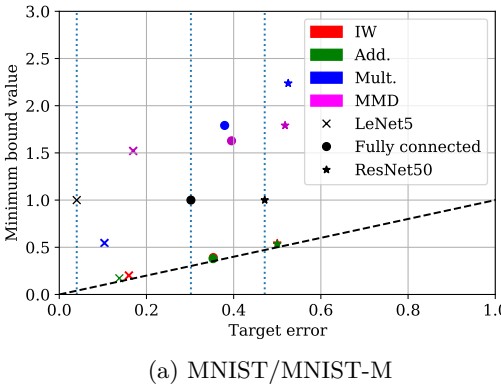 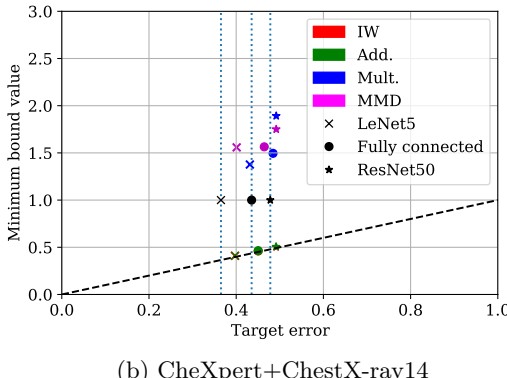

(a) MNIST/MNIST-M          (b) CheXpert+ChestX-ray14

Figure 5: An illustration of the minimum bound value achieved by each of the three architectures on both tasks. The lowest target error achieved is indicated by a black marker with a vertical dotted line through it.

disagreement will be low, using informed priors will make the $D_{\mathrm{KL}}$ small while using neural networks often lead to having a low source error. This will leave only the constant term, $\log \frac{1}{\delta}$ and the unobservable $\lambda_\rho$ terms and in those situations the bound might even be dominated by the unobservable term.

The multiplicative bound of (Germain et al., 2016) (**Mult**) suffers from the amplification of the source error $e_S$ and $D_{\mathrm{KL}}$ term by the factor $\beta_\infty$, and is generally larger than the **Add** and **IW** bounds. Conceptually, the **Mult** and **IW** bounds are similar, but in the former, the loss is multiplied uniformly by the largest weight. For tasks where certain inputs with high loss are more uncommon in the target domain than in the source domain, this is especially detrimental. The **MMD** bound is initially dominated by the MMD distance between inputs from the source and target domains, as shown in 4c, which is large and independent of the learned hypothesis. As such, this term cannot be reduced by optimization, without, for example, computing it in representation space (Long et al., 2015). With this approach, unobservable errors due to non-invertible representations must be accounted for (Johansson et al., 2019).

Experiments on using bounds for early stopping and model selection with different architectures yield the results seen in Figure 5. We can see that the errors achieved by terminating training at the smallest bound value (colored markers) do not coincide with the best-achieved target performance during training (denoted by the vertical dotted lines). Clearly, the bounds are not tight enough to do early stopping. This is a result of the sample generalization term $D_{\mathrm{KL}}$ increasing during training. For other analyses, this need not be the case. For larger architectures, the early-stopped models are closer to the best target models. If we instead look at the same figure again, but this time focus on utility for model selection we find something interesting. It seems that the bounds might be useful in this regard as they consistently have lower values for architectures/models which perform well. However, to be able to say this conclusively a more thorough study with different learning setups must be done. Both of the two previous observations should be contextualised with the fact that the domain shift terms are not dependent on the model as such, but amplify looseness in the case of the **Mult** and **IW** bounds. For the **MMD** and **Add** bounds, increased looseness during training is an artifact only of sample generalization.

As we can see in Figure 7a, when we vary the size of the images we give to the ResNet50 architecture we observe that the error seems to decrease for the larger image sizes. Although, the minimum bound value achieved does not seem to follow the same trend consistently. This is likely the result of the amount of epochs trained for both prior and posterior. In Figure 7b, we see that the choice of prior sample proportion $\alpha$ makes some change to the smallest bound achieved. We also see the minimum bound values grow for large values of $\alpha$, indicating that the remaining data is better spent calculating the bound than informing the prior in this case. We can also infer from Figure 1 that using no data to inform the prior is worse than using some. The overall shape is consistent with the results reported in Dziugaite et al. (2021, Figure 1).

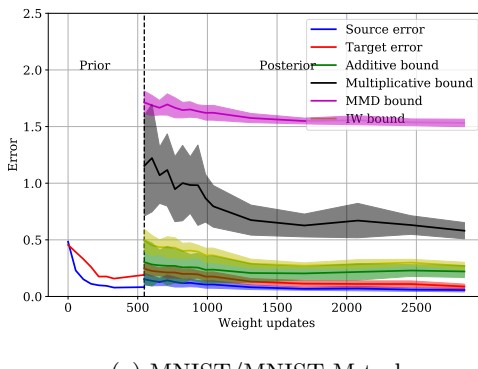 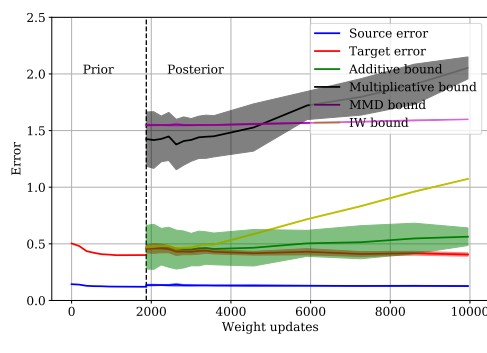

(a) MNIST/MNIST-M task                    (b) CheXpert+ChestX-ray14 task

Figure 6: Bounds evaluated at different points during training of the LeNet-5 architecture. $\alpha = 0.3$, $\sigma = 0.03$. The shaded areas represent one standard deviation.

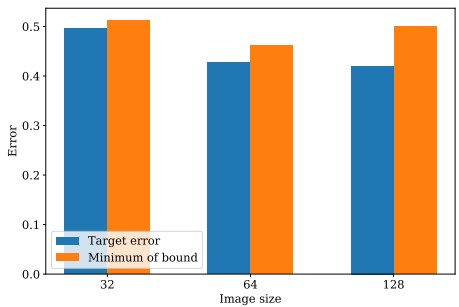 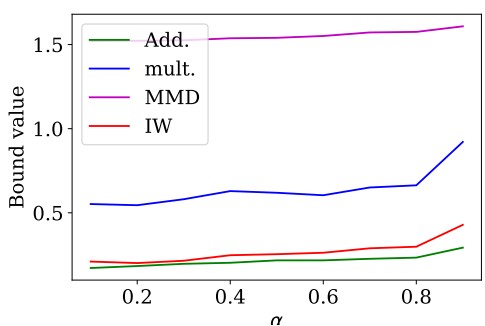

(a) Minimum bound value and target error on the X-ray task with different sizes of input images. Architecture used is ResNet50, $\alpha = 0.3$, $\sigma = 0.003$

(b) The minimum value of the bounds on the MNIST/MNIST-M task for different values of $\alpha$. The architecture used is LeNet-5.

Figure 7: a) show results of varying image sizes when training the ResNet50 network on the X-ray task. b) shows how the minimum value of the bound varies with $\alpha$.

## 5 Discussion

From our survey of the literature, it is clear that only a small handful of analyses of UDA generalization can be informative as practical bounds on target domain performance. The main obstacle for computing existing bounds is that they are vacuous or intractable to compute for the kinds of models which perform the best on common UDA benchmarks—deep neural networks. A potential remedy is the use of PAC-Bayes bounds, which perform well once they are applied with data-dependent priors; without this they are vacuous. In our experiments, the **Add** bound with the unobservable term is the tightest which is unsurprising given its dependence on target labels. Furthermore, we note that the application of importance weights also performs very well as the setting is sufficiently benign. As such we can say that in this setting we can achieve the desiderata of a tractably computable, tight bound using the **IW** bound. However, recall that the guarantee we get is on a distribution over classifiers and not on one specific classifier. It should be noted, however, that the **IW** bound can become vacuous in certain situations where the worst-case density ratio, $\beta_\infty$, is large and either the $D_{\mathrm{KL}}$ term or errors on underrepresented classes is large enough.

We found that the lowest value of the bounds achieved during training does not in general correspond to the best performing model on target. This tells us that these bounds are not useful metrics for early stopping. Further, the findings for using bound values for model selection are inconclusive, more experiments have to be conducted to answer this question satisfactorily. During training, we see that the dynamics are dominated by

the KL-divergence term, inherent to PAC-Bayes analysis, as training progresses. This reinforces our view that these bounds might be useful in getting performance estimates of methods at one particular point and not over several points during training if we do not have access to a large sample. This issue might be ameliorated by regularizing towards the prior during training, although this introduces yet another optimization as we now have to find the optimal regularization strength. In addition, it is not certain whether this will have any adverse effects on the final performance of the learned classifier.

A limitation of this work is that the bounds are cumbersome to compute and it is possible to do several optimizations in the process of producing the bounds. We have tried to do as few as possible in the name of practicality. We list some of the possible further optimizations in Appendix A for the reader's consideration. The impracticality of computing PAC-Bayes bounds is a known issue that has had some work done by Viallard et al. (2021) where they introduce an approach which would remove the computation of expectation over the posterior. In addition, in this work the computation of test errors have dominated the computation time. To produce the results one has to compute the predictions at least 50 times(5 pairs of sampled models from the posterior and 5 random seeds) for each datapoint in the bound for a single choice of prior. This will naturally consume increasing amounts of time with larger data sets.

Furthermore, the overlap assumption will not hold for all real-world applications. In fact, many of the benchmarks for algorithm development, such as the SVHN→MNIST task (Ganin et al., 2016) blatantly violate overlap, since images of house numbers and handwritten digits differ vastly in pixel space. Examples where overlap holds by definition include when the target domain represents a subpopulation of a larger population given by the source domain, e.g., women (target) among all patients (source) with a medical condition. Although an easy learning problem on its face, the optimal model in the full population may not be optimal for the subpopulation. Even when overlap is violated, many share the intuition that overlap may hold in a transformed space (Wu et al., 2019), representative of the core aspects of the problem—a digit is a digit, whether on a house or a postcard.

The strictness of the overlap assumption has been studied by D'Amour et al. (2021) where it was found that even for Gaussian distributions with insubstantial differences in mean parameters, overlap vanishes in high dimensions. Motivated by this fact we might wish to adopt relaxed versions of our assumptions or completely novel ones which still guarantee consistent estimation. A first step could be to require overlap only in a transformed space, not in the input space, like in Wu et al. (2019) or only requiring overlap in specific regions and leveraging assumptions on "closeness" in the other regions, as in Johansson et al. (2019). Further, task-specific assumptions are likely needed for a more complete description of out-of-distribution generalization. We mean task-specific in the sense that the assumptions will depend on the structure on the problem and the data-generating process (Hansen, 2008) or other approaches. Overcoming this gap is a important direction of future study.

Another limitation of this work is that the hypotheses do not optimize for adaptation to the target domain, which might be achieved through representation learning as in Ganin et al. (2016) or minimization of a weighted loss (Shimodaira, 2000). Our setting is representative for tasks where the target domain is unknown during training, but known when computing the bounds. Further, in this work we have assumed that we are able to estimate the importance weights exactly which may not be feasible in high-dimensional settings. In addition, there is no guarantee that the estimation error of the weights is small and thus even a small misestimation may have quite large implications for the resulting bound.

Future work regarding generalization bounds should preferably comment upon usefulness of their bound as a practical guarantee for performance, which is something that is often lacking. Ideally this would extend to explicit calculation if the bound is possible to compute. New bounds are often used as inspiration towards new algorithms which are hoped to result in more generalizable models. However, this is seldom guaranteed by theory and verified only in limited settings empirically.

Our results offer indications for how to obtain tractable and tight bounds for neural networks used in UDA tasks with available tools. If overlap can be assumed to hold, then use the **IW** bound, estimate importance weights using density estimation (Sugiyama et al., 2012) or probabilistic classifiers and apply data-dependent priors. The amount of data to use and how long to train your prior etc. are all task dependent and thus some engineering is necessary to pick optimal values. If this cannot be assumed, the most promising approach

to get bounds which fulfill our desiderata in this case would be to use the **MMD** bound as this does not technically rely on overlap and is tractable to compute for neural networks. This relies on the added assumptions of the pointwise loss being bounded by a function in the associated reproducing-kernel Hilbert space, which may or may not hold. The nature of this assumption makes it less useful since no test for this is available absent overlap and is similar in nature to assuming that joint optimal error is small. However, if the function under estimation is believed to be smooth, the assumption is more plausible. In conclusion, it is clear that the general case demands new research, and alternative, task-specific assumptions, to allow tight performance guarantees for realistic problems. In either setting, we conjecture that the tightest bounds will be coupled to the training procedure.

## Acknowledgements

This work was supported in part by the Wallenberg AI, Autonomous Systems and Software Program (WASP) funded by the Knut and Alice Wallenberg Foundation.

The computations and data handling were enabled by resources provided by the Swedish National Infrastructure for Computing (SNIC) at Chalmers Centre for Computational Science and Engineering (C3SE) partially funded by the Swedish Research Council through grant agreement no. 2018-05973. Mikael Öhman at C3SE is acknowledged for his assistance concerning technical and implementation aspects in making the code run on the C3SE resources.

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

## A    Experimental details

The experiments were carried out with a modified version of LeNet-5 due to Zhou et al. (2019), a 1024-600-600-2 fully connected network similar to the one used in Rivasplata et al. (2020) and a ResNet50 architecture (He et al., 2016) as mentioned before. Since the 0-1 loss is not differentiable we substitute it with the binary cross entropy loss which is and provides a tight upper bound on the 0-1 loss. We train with SGD with momentum 0.95 as the optimizer with a batch size of 128. The learning rate was chosen to be $3 \times 10^{-3}$ for the LeNet-5 and fully connected architectures while for ResNet50 $3 \times 10^{-4}$ was used. The images from MNIST and MNIST-M were padded with zeros to $32 \times 32$ images. This was done to be able to use them with the ResNet50V2 implementation in Tensorflow which does not support smaller image sizes.

The training procedure went as follows. We load the data and construct our source and target. The source data is then split into an $\alpha$-fraction $S_\alpha$ which is used to train the prior network on for $|S_\alpha|/b$ iterations, where $b$ is the batch size, to get an informed prior. This is then used as the starting point when training the posterior which is done on all the available data. During training of the posterior network we save 10 network weights during the first epoch and then at the end of every subsequent epoch until termination. We terminate the training of the posterior network when we have trained 5 epochs.

When training the posterior is terminated we save the weights and proceed with the computation of the bound. In contrast with Dziugaite et al. (2021) we not only consider the bound at the point of termination but also at previous points during training. This is done with the goal of gaining an understanding of the bounds' behaviour during training.

We assume that our prior can be modeled by an isotropic gaussian akin to earlier work, this is done to get an easily computable closed form expression of the KL divergence. To pick a good value of the $\sigma$ parameter one could sweep over some range of values and then use a union bound argument to be able to pick the best result with only a small penalty to the bound. We do not do such optimization and simply pick a value.

We perform a small optimization step when determining the free parameters of the bound. For the bounds from Germain et al. (2020), i.e. $a, b, \gamma$ and $\omega$, we iterate over values in the range $\{1 \times 10^{-3}, 5 \times 10^{-3}, 1 \times 10^{-2}, \dots, 5 \times 10^4, 1 \times 10^5\}$ for both free parameters and pick the combination which yields the lowest bound. For the **MMD** and **IW** bounds we pick $\gamma$ from the range $\{1 \times 10^{-3}, 5 \times 10^{-3}, 1 \times 10^{-2}, 5 \times 10^{-2}, 1 \times 10^{-1}, 5 \times 10^{-1}, 9.9 \times 10^{-1}\}$, choosing the one which yields the lowest bound.

### A.1    Possible optimization when producing the bounds

When computing these bounds there are a lot of different parameter and hyperparameter choices to make, many of which can be optimized. We first have to train at least one model(depending on how many values of $\alpha$ to consider) with all parameter choices that entails. Then we sample models according to whatever the posterior distribution is; the amount depending on how well we want to estimate the expectation over the posterior. All PAC-Bayes bounds contain some sort of parameter which is free to choose and we must do a at least a rudimentary parameter search to arrive at a good bound. In addition to all these choices of parameters we can of course optimise these bounds even further. Some that we did not perform for this work are: Optimise the representation for smaller MMD, L2 regularisation towards the prior for each specific parameter set (also entails finding the optimal regularization strength) and perform even finer grid searches for the optimal bound parameters to name just a few.

## B    Importance weights and how to derive them

So assume that we do a mixing of two data sets (let's call them 0 and 1) to form two domains. We want to derive the way we should calculate and subsequently use the importance weights for this situation. We will do this first for the CXR task. In this task the underlying label set is multi-label and as such we need to make it into categorical variables before calculating and applying weights. We achieve this by making the problem into a binary classification problem where we try to predict if there is a finding or not. From this point we may calculate the importance weights as follows:

$$w = \frac{T(x,y)}{S(x,y)} = \frac{T(x|y)T(y)}{S(x|y)S(y)} = \frac{T(x|y)T(y)}{(S(x|y,D=1)S(D=1|y) + S(x|y,D=0)S(D=0|y))S(y)}$$

If we now assume that we have only no examples from data set 0 in the target as in the CXR task then we have the following

$$w = \frac{T(x|y)T(y)}{(S(x|y,D=1)S(D=1|y) + \underbrace{S(x|y,D=0)S(D=0|y)}_{=0,\ \text{as T(y)=0 when this is non-zero}})S(y)} = \frac{T(x|y)T(y)}{(S(x|y,D=1)S(D=1|y)S(y)}$$

Now we note that $T(x|y)$ and $S(x|y,D=1)$ cancel as the conditional distribution of these should be the same as we mixed uniformly over the initial label and T(x|y,D=1)=T(x|y). We are thus left with

$$w(y) = \frac{T(y)}{S(y)S(D=1|y)} = \frac{\#\text{examples with label y in T}}{\#T} \Big/ \frac{\#\text{examples with label y in S which come from data set 1}}{\#S}$$

Through this argument we can see that the final importance weight is in the case where we use 20% of the images from data set 1, which will become the target, to mix with data set 0 to become the source. Assume that the initial amount from data set 1 is $m_1$.

$$w = \frac{\#S}{\#T} \cdot \frac{\#\text{examples w/ label y in T}}{\#\text{examples w/ label y in S which come from data set 1}} = \frac{\#S}{\#T}\frac{0.8m_1}{0.2m_1} = 4\frac{\#S}{\#T}$$

We can do the same type of argument for the MNIST/MNIST-M mix. There we have a more balanced data set where the classes are evenly distributed in amount across source and target.

$$w = \frac{T(x,y)}{S(x,y)} = \frac{T(x|y,D=1)T(D=1|y) + T(x|y,D=0)T(D=0|y))T(y)}{(S(x|y,D=1)S(D=1|y) + S(x|y,D=0)S(D=0|y))S(y)}$$

Since the labels are balanced between the data sets $\frac{T(y)}{S(y)} = 1$. Since we have mixed the datapoints for each label in a uniform fashion we know what $\frac{T(x|y,D=0)}{S(x|y,D=0)} = 1$ for every label. As such we can calculate the weight as

$$w(x|y,D=0) = \frac{\#\text{examples w/ label y in T which come from data set 0}}{\#\text{examples w/ label y in S which come from data set 0}}$$

and similar for datapoints from the other data set.

## C    Additional results

### C.1    Constituent parts of bounds

### C.2    Best bounds achieved for different prior sample proportions

### C.3    Bound during training

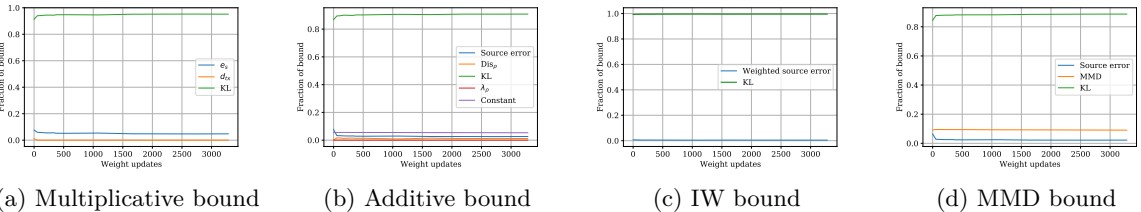

(a) Multiplicative bound      (b) Additive bound      (c) IW bound      (d) MMD bound

Figure 8: An illustration of constituent parts of each of the four bounds with the fully connected architecture on the MNIST mixture task. $\alpha = 0$

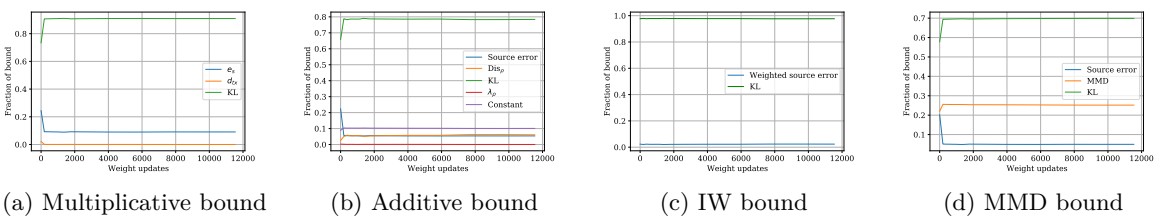

(a) Multiplicative bound      (b) Additive bound      (c) IW bound      (d) MMD bound

Figure 9: An illustration of constituent parts of each of the four bounds with the fully connected architecture on the X-ray task. $\alpha = 0$

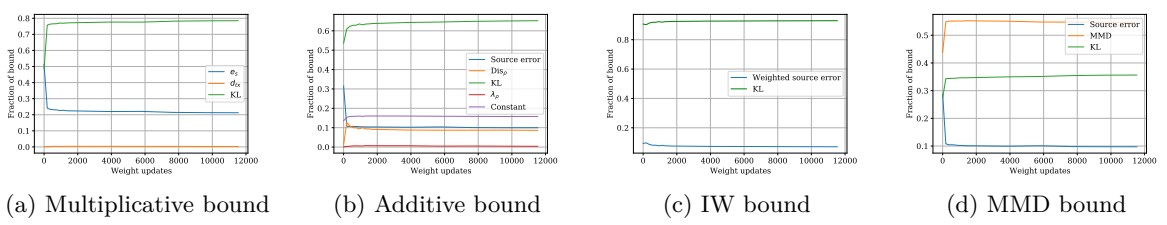

(a) Multiplicative bound      (b) Additive bound      (c) IW bound      (d) MMD bound

Figure 10: An illustration of constituent parts of each of the four bounds with the LeNet-5 architecture on the X-ray task. $\alpha = 0$

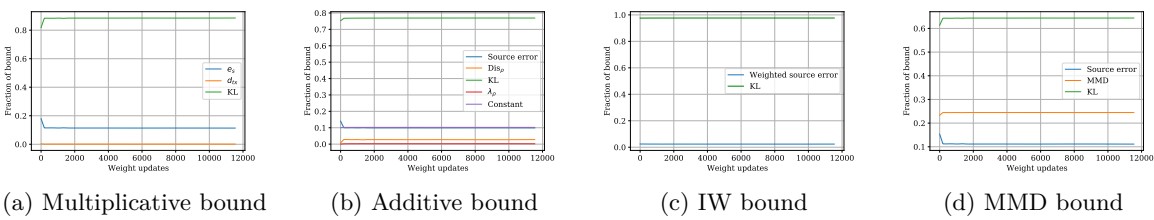

(a) Multiplicative bound      (b) Additive bound      (c) IW bound      (d) MMD bound

Figure 11: An illustration of constituent parts of each of the four bounds with the ResNet50 architecture on the X-ray task. $\alpha = 0$

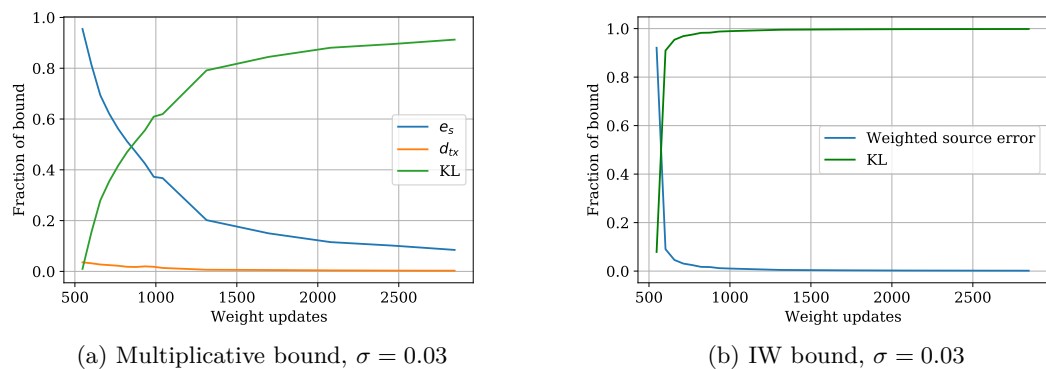

(a) Multiplicative bound, $\sigma = 0.03$        (b) IW bound, $\sigma = 0.03$

Figure 12: An illustration of constituent parts of each of the four bounds with the fully connected architecture on the MNIST mixture task. $\alpha = 0.3$

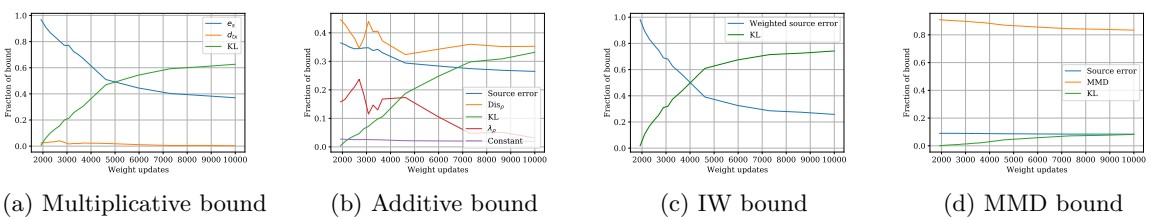

(a) Multiplicative bound     (b) Additive bound     (c) IW bound     (d) MMD bound

Figure 13: An illustration of constituent parts of each of the four bounds with the fully connected architecture on the X-ray task. $\alpha = 0.3$, $\sigma = 0.03$

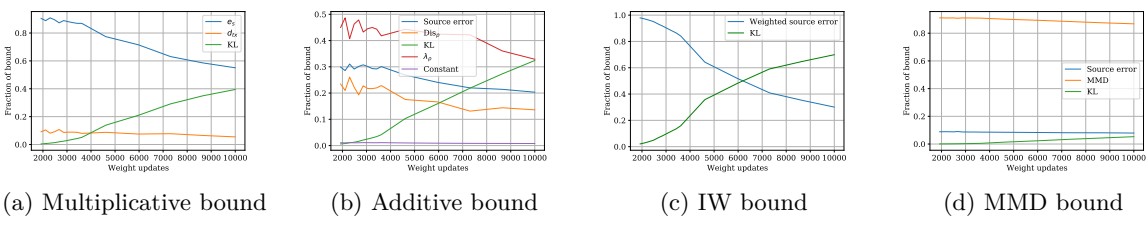

(a) Multiplicative bound     (b) Additive bound     (c) IW bound     (d) MMD bound

Figure 14: An illustration of constituent parts of each of the four bounds with the LeNet-5 architecture on the X-ray task. $\alpha = 0.3$, $\sigma = 0.03$

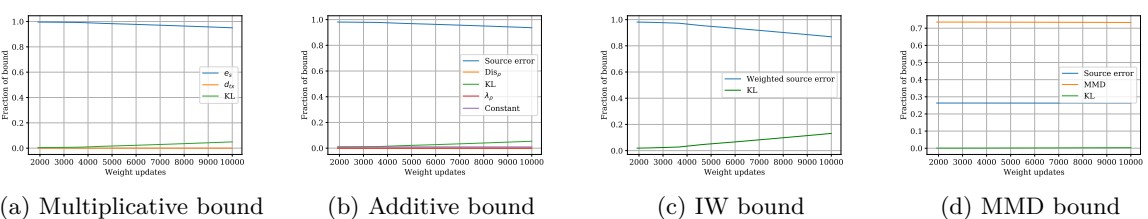

(a) Multiplicative bound     (b) Additive bound     (c) IW bound     (d) MMD bound

Figure 15: An illustration of constituent parts of each of the four bounds with the ResNet50 architecture on the X-ray task. $\alpha = 0.3$, $\sigma = 0.03$

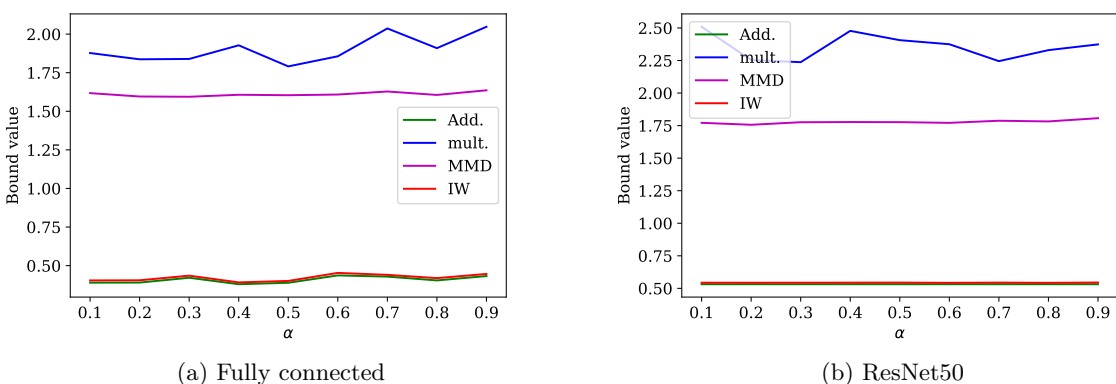

(a) Fully connected

(b) ResNet50

Figure 16: The minimum value of the bound on the MNIST mixture task for different values of $\alpha$.

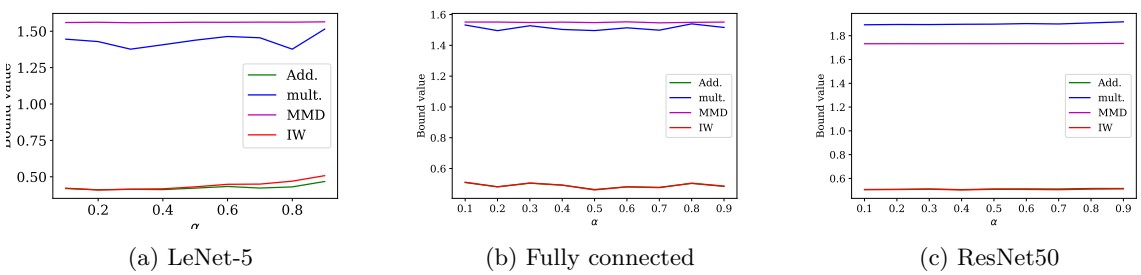

(a) LeNet-5

(b) Fully connected

(c) ResNet50

Figure 17: The minimum value of the bounds on the X-ray task for different values of $\alpha$.

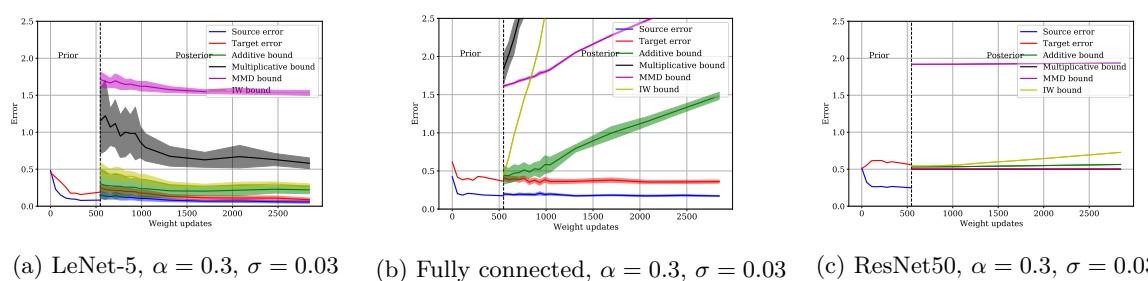

(a) LeNet-5, $\alpha = 0.3$, $\sigma = 0.03$

(b) Fully connected, $\alpha = 0.3$, $\sigma = 0.03$

(c) ResNet50, $\alpha = 0.3$, $\sigma = 0.03$

Figure 18: The evolution of the bounds during training on the MNIST mixture task when we use 30% of our sample to inform the prior.

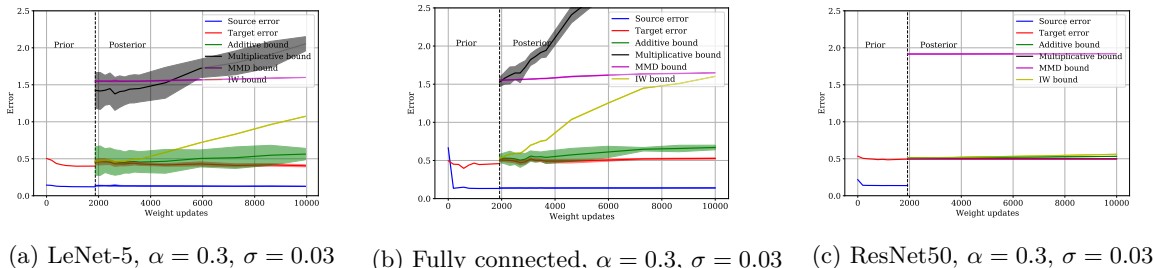

(a) LeNet-5, $\alpha = 0.3$, $\sigma = 0.03$

(b) Fully connected, $\alpha = 0.3$, $\sigma = 0.03$

(c) ResNet50, $\alpha = 0.3$, $\sigma = 0.03$

Figure 19: The evolution of the bounds during training on the X-ray task when we use 30% of our sample to inform the prior.

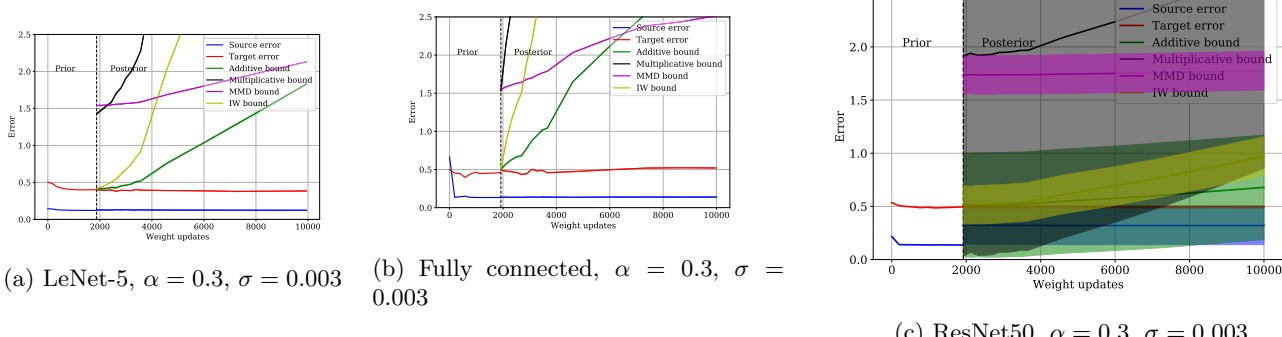

(a) LeNet-5, $\alpha = 0.3$, $\sigma = 0.003$

(b) Fully connected, $\alpha = 0.3$, $\sigma = 0.003$

(c) ResNet50, $\alpha = 0.3$, $\sigma = 0.003$

Figure 20: The evolution of the bounds during training on the X-ray task when we use 30% of our sample to inform the prior.

