# OpenReview forum: "Practicality of generalization guarantees for unsupervised domain adaptation with neural networks"
_TMLR — Accepted by TMLR_

### Review · Reviewer_TMKi · 2022-08-09

**Summary Of Contributions:**

The authors are interested in the tightness, estimability, and computability of generalization bounds for domain adaptation. Their study is mainly motivated by the failure of most of the bounds in the literature to satisfy the three desiderata for neural networks, while the latter constitute the bulk of domain adaptation algorithm implementations. They restrict the study to the benchmark configuration where the covariate shift assumption holds and the source domain's support includes target's. To tackle such limitations, they advocate relying on PAC-Bayes analysis with a data-dependent prior. On the experimental side, they consider 4 generalization bounds and illustrate the advantage of data-dependent priors over uninformative ones. Finally, the limitations and possible extensions of the method are discussed.

**Broader Impact Concerns:**

This paper is concerned with a general aspect of DA theory with no focus on a particular application. Nevertheless, one can point out for example to the consequences of deploying domain adaptation methods on medical tasks, which might have important risks in case of failure of adaptation.

**Requested Changes:**

# Main points
According to the weaknesses pointed out in the previous section, I kindly request the following changes/additions from the authors:
* Including the lacking literature aspects, i.e. bounds based on asymmetrically relaxed divergences, localization, and f-divergences. In particular, there should be a theoretical discussion on the tightness of such bounds.
* Since the experiments concern several seeds, including the deviation from the mean can be more informative on the behavior of the bounds.
* Either taking into account the estimation of importance weights, or clearly stating that it is assumed that they are known.
* Stating that the importance weights are assumed to be uniformly bounded.
* Adding a discussion on how $K$ behaves for a neural network.

# Minor points
* In figure 5, a logarithmic scale figure for values close to the target error, and adding a grid, might facilitate reading.
* In the discussion section, it is mentioned "The main obstacle for using existing bounds is that they ... ". I would say "computing" rather than "using", are such bounds are used to propose DA algorithms in several cases.

**Strengths And Weaknesses:**

The paper is well written and addresses an aspect of domain adaptation that is almost absent in the DA literature, namely the tightness of generalization bounds. However, it misses some aspects of the DA literature.
# Strengths:
* The interest in the value of DA bounds, which is different from the mainstream role of inspiring algorithms: this is an important aspect as most of these bounds constitute an inspiration for designing DA algorithms.
* The experimental procedure is well detailed in the main paper, and additional details are provided in the appendix.
* Several limitations of the work are discussed in detail.
# Weaknesses:
* Some literature aspects of DA bounds are lacking, namely:
  * Asymmetrically relaxed discrepancies [3], where the importance weight is only required to be bounded in order to guarantee good adaptation, instead of requiring equality of distributions.
  * Localized discrepancies [1,2], where the restriction of the supremum defining the divergence to a proper subset of the hypothesis space yields significantly lower divergence measures as illustrated in [1].
  * Other important DA contributions with bounds leading to algorithms such as [4].
* On importance weighting and the IW bound (Corollary 1):
  * The IW bound proposed in the paper can become vacuous if $\beta_\infty = \infty$. This can happen even if the source domain's support completely includes the target's.
  * In the IW bound, the estimation error of the weights is not taken into account, i.e. it assumes perfect computation of the importance weights $w(x)$. This goes against the spirit of the paper concerned with the tightness of generalization bounds.

* The MMD bound contains $K$, a parameter of the kernel that controls the estimation error of the MMD. However, no discussion is provided on its behavior for neural networks, whereas a main motivation of the paper is the vacuity of classic generalization bounds in the case of neural networks, notably those relying on the VC dimension.

# References
[1] Zhang, Yuchen, et al. "On localized discrepancy for domain adaptation." arXiv preprint arXiv:2008.06242 (2020).

[2] Cortes, Corinna, Mehryar Mohri, and Andrés Munoz Medina. "Adaptation based on generalized discrepancy." The Journal of Machine Learning Research 20.1 (2019): 1-30.

[3] Wu, Yifan, et al. "Domain adaptation with asymmetrically-relaxed distribution alignment." International conference on machine learning. PMLR, 2019.

[4] Acuna, David, et al. "f-domain adversarial learning: Theory and algorithms." International Conference on Machine Learning. PMLR, 2021.

---

> ### Author Response · Authors · 2022-09-06
> **Response to Reviewer TMKi**
>
> We thank the reviewer for their comments and helpful references to related work. The comments below are regarding the requested changes.
>
> Regarding the importance weighting bound we have stated explicitly in section 3 that we assume that they are uniformly bounded and that we can calculate them exactly, based on knowledge of the data set construction, for simplicity.
>
> We have added standard deviations to the bounds in figure 6 and similar figures in the appendix.
>
> We thank the reviewer for reminding us of the works using localized discrepancies. We have added a discussion on these to the second section and a short note of the intractability of computing the discrepancy for a subset of promising classifiers when these are deep neural networks. This disqualifies them from consideration in this work as we study practically computable bounds.
>
> Further on the point of lacking literature coverage. Paper [3] is primarily focused on achieving a bound which might be successfully minimised (in part) by methods like DANN. It thus uses a very specific construction which is unique in the sense that it requires a lot of assumptions on the structure of the domains under a representation mapping and how they are related. We have added a note at the end of section 2 exemplifying this as a direction for future work. However, we do not consider this work along with other representation learning approaches for reasons stated at the end of section 2.
>
> The last paper, [4], also focuses on constructions that can be effectively controlled by adversarial learning. We have added it to the overview section. However, the resulting bounds also contain the joint optimal error parameter $\lambda$ which disqualifies it similar to many other works considered in section 2.
>
> Regarding the boundedness of the kernel. While we have not explicitly mentioned it in the paper, $K$ is bounded by 1 for our choice of kernel, the Gaussian RBF-kernel, irrespective of model. Hence, we can compute all parts of the bound.  However, making any concrete statement about the variation in tigher bounds on $K$ for neural networks is very hard. Generally speaking, the value might be high or low depending on the type of data considered, what kernel one chooses to work with etc. The kernel value itself is not directly influenced by the model choice, other than through the assumption that the loss can be pointwise bounded by a function in the RKHS given by $k$.

---

### Review · Reviewer_iFWv · 2022-08-15

**Summary Of Contributions:**

The paper investigates the utility of generalisation bounds for the unsupervised domain adaptation setting with two assumptions of the type of distribution shift: covariate-only shift, and $\text{supp}(\mathcal{T}) \subseteq \text{supp}(\mathcal{S})$. A review of existing bounds is provided, where each is identified as being practically estimable/computable. Four PAC-Bayes bounds are identified/derived, and compared experimentally. The empirical results determine that these bounds are sometimes non-vacuous.

**Broader Impact Concerns:**

There are no broader impact concerns.

**Requested Changes:**

In my view, the work does not need any significant changes to justify acceptance. Some additions the authors may consider are including sample efficiency experiments, and providing a brief introduction to the MMD tools used for one of the bounds.

**Strengths And Weaknesses:**

The paper is well-written and easy to follow. The scope of the problem considered in this paper is clearly described, and the limitations of the results are also identified. I also appreciate the thorough review of existing bounds, and the annotations for each bound provided in Table 1.

The technical content is interesting. The theoretical contributions are, although "easy", still quite a useful combination of existing work that address an important gap in the literature. I found the idea of combining the data-dependent PAC-Bayes method from Dziugaite et al. and the importance weighting/MMD tools to be interesting.

The empirical results are also quite interesting. I especially liked the decomposition of the bounds into their constituent terms to determine where improvements can be made, and the investigation into the utility of using these bounds for model selection.

Something that would have been good to include is a plot (or several) demonstrating sample efficiency. I.e., as the total number of training examples is increased, how tight do the bounds get?

---

> ### Author Response · Authors · 2022-09-06
> **Response to Reviewer iFWv**
>
> We thank the reviewer for taking their time to review our work. We are glad to hear that the reviewer finds our extensions of previous work interesting. We share the sentiment.
>
> We have added a short introduction and definition of the MMD.
>
> Concerning the effect of sample size. We have investigated the effect of simply increasing the amount of datapoints available when calculating which of course tightens the bounds. However, the question is what exactly increasing the sample means. If we constrain the total amount of datapoints, it is more difficult to disentangle effects as both the posterior learned and the bound calculation are affected.
>
> If the reviewer refers to just constraining the amount used for the bound it is simply a exercise of increasing $m$ in the bounds. This results in the IW and Add bounds becoming increasingly tight as they converge to the target risk. The Mult. and MMD bounds will still have the $\beta_\infty$ and $MMD$ terms respectively which will not allow them to become fully tight unless they are 1 and 0 respectively.

---

### Review · Reviewer_vsaP · 2022-08-31

**Summary Of Contributions:**

This paper provides a PAC-Bayesian analysis of domain adaptation on image classification tasks, where most generalization bounds are vacuous. The authors incorporate PAC-Bayes theory into several bounds and use importance weighting to obtain tighter guarantees. They evaluate the bounds empirically on two tasks satisfying the covariate shift and domain overlap assumptions.

**Broader Impact Concerns:**

N.A.

**Requested Changes:**

Overall, I think that this is a good work that is relevant to the TMLR community. I hope that the authors can elaborate on the points mentioned above.

**Strengths And Weaknesses:**

Strengths: The paper is well written, well organized and provides a good rationale for their motivation with a detailed review of the literature. Although the idea of using PAC-Bayes in domain adaptation is not new, the authors investigate this idea in details in the current paper, and give interesting results on the topic. They examine the tightness of existing bounds when using deep neural networks, identify those which could be estimable and tractably computable, and show that it is possible to get tight bounds when using standard PAC-Bayesian optimization techniques. The main results and the empirical study are of great interest.

Weaknesses: A general criticism that I have is that some points in the paper could have been investigated more thoroughly. This is the case, for instance, of the computational practicality of the bounds. Also, the usefulness of the bounds in model selection, which is a huge interest of PAC-Bayes bounds, is inconclusive here and left for future research. I also have some minor concerns on the assumptions of the paper. As noticed by the authors, the overlap assumption is quite strong and is a huge limitation of the current work. I would have appreciated more discussion on this point. Furthermore, the other MMD bound is based on the assumption that the pointwise loss is bounded by a function in the RKHS, which is much less natural and intuitive. The authors could have detailed how the two bounds fit together and why these assumptions might (if possible) be complementary.

---

> ### Author Response · Authors · 2022-09-06
> **Response to Reviewer vsaP**
>
> We thank the reviewer for reviewing our work and their insightful comments.
>
> Concerning practicality of computation, we considered a restricted case where it is at least possible to accurately estimate the quantities in the bounds. The practicality of training a large number of models for this purpose depends on the size of the networks and the nature of the tasks. We have added a brief discussion of this in the limitation section of the Discussion.
>
> We concur that the use of the PAC-Bayes bounds for model selection is an interesting topic. However, we found that investigating this in a more complete manner would have increased the amount of work substantially while not being directly related to the computation and comparison of the bounds, which is our primary focus. Therefore we chose to leave such an investigation to future work.
>
> We agree with the reviewers points that assuming overlapping support is likely too strong of an assumption to be empirically viable and the assumption required for the MMD bound to hold is fairly opaque. The MMD assumption does not rely on overlap, but whether it is satisfied depends on the behavior of loss functions outside of the support of the labeled data. As such, it is not verifiable without overlap. If there is reason to believe that the function under estimation is smooth, the assumption is more plausible but should nevertheless be used with care. More work is needed to intuit for which tasks and models this is a reasonable assumption and when other ones are needed. We expanded the discussion on the intuition of overlap and the loss assumption in the discussion section.

---

> > ### Comment · Reviewer_vsaP · 2022-09-19
> > **Response**
> >
> > I would like to thank the authors for their detailed response and for their fine work. I enjoyed reading it, and although the ideas presented here are not completely new, they are nicely combined. I think the article deserves to be accepted.

---

### Author Response · Authors · 2022-09-06
**Revision submitted**

We have submitted a revision of the paper which has incorporated changes requested by the reviewers. The edits made are marked with blue text. Best, The authors

---

### Decision · Action_Editors · 2022-10-03

**Recommendation:** Accept with minor revision

**Comment:**

This paper provides a PAC-Bayesian analysis of domain adaptation on image classification tasks. The key technique is the construction of PAC-Bayes type bounds with importance weighting for improving tightness. Some experiments on image classification are conducted as to verify the proposed bounds.

Although the use of PAC-Bayes is not new in domain adaptation, reviewers found the paper interesting and they agree the contributions made in this paper are worth publishing.

I recommend for "accept with minor revision" to give the authors an opportunity to better reflect on the reviews. Specifically, the authors may want to further clarify the following points in the final version:
1. the usefulness of their proposed bounds for model selection (as the authors claimed the bounds are non-vacuous);
2. assumptions of the theoretical results, whether they can be further relaxed or not;
3. the IW bound: whether it might become vacuous under some circumstances, also the estimation error of the importance weights in practice.

**Audience:**

This paper is about domain adaptation and using techniques from PAC-Bayes. So it would be of interest to at least these communities.

**Claims And Evidence:**

A good paper mainly focusing on the theoretical analysis but also with some experimental support. Included a table summarising historical results which is welcomed by reviewers.